# Environmental attitude and affective-motivational beliefs towards sustainability of secondary school children in Germany and their associations with gender, age, school type, socio-economic status and time spent in nature

**Charlotta Bucht**[1]*, **Joachim Bachner**[1], **Sarah Spengler**[2]

**1** Didactics in Sport and Health, TUM School of Medicine and Health, Technical University of Munich, Munich, Germany, **2** Sports Science, University of Konstanz, Konstanz, Germany

* lotta.bucht@tum.de

## Abstract

There are warnings that human actions will lead to irreversible environmental damage if they continue at their current pace and scale. With regard to individual aspects, a pro-environmental attitude and positive affective-motivational beliefs towards sustainability represent fundamentals for a more sustainable future. However, there is little data regarding these constructs and relevant explanatory factors, especially with regard to young people. We examined environmental attitude (two-dimensional: utilization and preservation) and affective-motivational beliefs towards sustainability with regard to gender, age, socio-economic status, school type and time spent in nature in 484 adolescents (11–14 years) living in German cities by means of univariate and multiple regression analyses. Mean values were high in preservation and affective-motivational beliefs towards sustainability, and relatively low in utilization. Female adolescents had higher values compared to male students in affective-motivational beliefs towards sustainability. Age did not play a substantial role. Although being strongly correlated with each other, school type and socio-economic status each exhibited positive associations to environmental attitude and affective-motivational beliefs towards sustainability. Furthermore, multiple regression analyses identified time spent in nature as a significant predictor of incremental value, suggesting a tentative recommendation to spend at least half an hour per week in nature in order to promote positive attitudes towards the environment and sustainability. In sum, special needs for topic-related education seem to exist for male students, students with lower formal level of education, students with a lower socio-economic status and students who spend little time in nature.

**Data Availability Statement:** All relevant data are within the paper and its Supporting Information files.

**Funding:** This research was funded by German Federal Ministry for the Environment, Nature Conservation, Nuclear Safety and Consumer Protection, grant number 67KF0092. SS and CB received this funding for their work in the "Klima bewegt!" study. The funders had no role in study design, data collection and analysis, decision to publish, or preparation of the manuscript.

**Competing interests:** The authors have declared that no competing interests exist.

# Introduction

Our planet is facing environmental problems, like climate change or land-system change [1], as a result of human actions [2–4]. Rockström [5] developed the concept of planetary boundaries. According to these predictive calculations, critical boundaries for some parameters, like the ones for climate change or biogeochemical flows, have already been exceeded. This means that the changes in these parameters cannot be reversed [1, 6, 7]. At the same time, there are more than two hundred eighty million migrants worldwide, making up 3.6% of the global population, as a result of natural disasters and conflicts [8]. Politicians, scientists and society agree that environmental and sustainability issues are the greatest challenge of the decade [9–11]. Without a drastic change towards achieving sustainable development, the consequences of environmental change will have a much larger impact on Earth and its societies than today [7].

The term environment is defined as "the external conditions, resources, stimuli, etc., with which an organism interacts" and comprises different types of environment [12]. In this paper, we always refer to the meaning of the natural environment, which includes a complex interplay of human beings and other living species, like animals or plants, and their natural surroundings, e.g. climate or natural resources [13].

Sustainability has three dimensions. The economic, social and environmental dimensions influence each other and can only exist together [14]. Therefore, sustainability means that all three dimensions should be considered simultaneously and equally [14]. Sartori and colleagues [15] state that sustainability requires sustainable development. Aiming to drive sustainable development globally, the Agenda 2030 was launched in 2015 and includes 17 sustainable development goals (SDGs) [16]. These goals were developed so that there are tangible goals global society can work towards to create a viable future for all. Different approaches are needed to promote sustainable development [16]. This requires, for example, constructive and appropriate political decisions and technological progress, but also increased awareness among the population. This includes the awareness for justice and injustice, environmental protection and environmental use and also the ability to behave in a sustainable way [16, 17]. Today's young generation deserves particular consideration in this, because young people are faced to live with the consequences of environmental change and can still be reached well with educational programs. Additionally, they are currently the loudest age group in demanding change in politics and in the behavior of society [10, 18–20]. Thirty-one percent of young respondents in Germany say that environmental and climate protection are current important problems in society and politics [21]. Seventy-nine percent of Europeans between the ages of 15 and 24 believe that their national government is not doing enough to tackle climate change [22].

Problem awareness and attitudes towards sustainable development of the younger generation are already shaped by school education. In Germany, various school subjects, such as geography, biology, chemistry or politics, address environmental and sustainability topics (see, for example [23, 24]). Accordingly, it can be assumed that many students in Germany have some knowledge of topics such as "climate change", "education for sustainable development", "urbanization and urban development", "animals and plants", "agriculture and food" or "energy" [23, 24]. It is promising to focus on environment and sustainable development topics in secondary level 1 (when most children are between 11–16 years old), because the period between nine and 14 years of age represents a highly sensitive period for developing attitudes toward the environment [25].

Environmental knowledge can be divided into three types of knowledge: system-, action-related and effectiveness knowledge [26]. While system-related knowledge, which refers to basic knowledge, has no influence on environmental behavior, some studies show that action-related knowledge and effectiveness knowledge often have an influence on behavior [26]. Basic

knowledge about environmental topics alone is often not sufficient to cause behavioral change, for this reason, attitudes are extremely important for shaping personal behaviors. An attitude is the positive or negative evaluation of an object, person, group, issue or concept. Attitudes "provide summary evaluations of target objects and are often assumed to be derived from specific beliefs, emotions, and past behaviors associated with those objects" [27]. They can therefore be viewed as one of the indispensable piers of the bridge that leads from knowledge to behavior [28, 29].

According to Schick [30], environmental attitude can be viewed from a functional and a structural perspective. Functionally, environmental attitude can help to plan an action with an awareness of one's responsibility toward the environment. The structure includes affective (e.g. emotion), cognitive (e.g. knowledge), and conative (e.g. intention) dimensions [30, 31]. Roczen and colleagues [28] were the first ones to test the pro-environmental competence model from Kaiser and colleagues [32] with adolescents. It shows that environmental attitude, i.e. attitude toward nature, can be understood as the most important factor in motivating pro-environmental behavior. Additionally, environmental attitude can provide a link between environmental knowledge and resulting behavior [28, 29]. It is important to mention that environmental attitude must be considered as a two-dimensional construct comprising utilization (UTL) and preservation (PRE) [33, 34]. The PRE factor represents an individual's tendency to protect the environment, whereas the UTL factor indicates the tendency of a person to exploit the environment [34]. These two factors do not represent the positive and negative side of the same unidimensional construct, which is illustrated by the fact that both tendencies can exist in the same person. As an example, such a person would want to protect the environment, but at the same time accepts environmental exploitation because they are convinced that innovative technological progress can balance the exploitation of the environment.

If we look at sustainability attitudes in scientific literature, we find that there are only a few mentions of the term attitudes and often in combination with sustainability competence. In the field of sustainability, the terminology regarding attitudes revolves around competence, which follows a holistic approach. Waltner and colleagues [35] have developed a 3-level competence model. All levels have goal dimensions in cognitive, affective-motivational and behavioral areas. The affective-motivational goal dimension is very close to sustainability attitude in the respective levels [35]. The affective-motivational dimension of sustainability competence involves affect-related competencies, needs and motivation. These include, for example, attitudes, sense of responsibility or convictions [35].

Several aspects can influence environmental attitude and affective-motivational beliefs towards sustainability (AMBTS) [35, 36]. In the following, we will focus on gender, age, school type, socio-economic status (SES) and nature experiences.

Regarding environmental attitude, gender differences were not found in a study by Liefländer and Bogner [37]. There are further studies that show no or only small gender differences in environmental attitude [38, 39]. Eagles and Demare [40] defined moralistic attitudes as a "primary concern for the right and wrong treatment of the environment, with strong opposition to exploitation of or cruelty to the environment". Data of children living in Canada showed that girls (aged ten to eleven years) exhibited slightly higher moralistic attitudes toward the environment than boys [40].

Liefländer and Bogner [37] have shown that pro-environmental attitude in children of nine to ten years of age differ significantly from the older students aged eleven to 13 years. The younger children had higher values in PRE and lower values in UTL, which indicates a more positive attitude toward the environment [37]. Krettenauer [41] and other studies have confirmed the hypothesis that younger people have higher pro-environmental values [38, 42, 43].

In an American study, people with a higher education level had higher scores in environmental attitude [42]. The study by Péer and colleagues [44] has shown a positive relationship between students' environmental knowledge and their mothers' education. Simultaneously, although not statistically significant, there is a trend indicating that their mothers' education level also has an impact on students' environmental attitude [44]. No scientific literature directly related to the association between children's SES and their environmental attitude was found.

Regarding AMBTS, scientific studies are still rare [45]. In western countries, girls mostly reached higher scores in sustainability values than boys [35, 46, 47]. In Waltner et al. [35], it can be seen that values worsened with increasing age. To date, there are no studies that concentrate on the associations between AMBTS and school type, SES or nature experiences.

Regarding nature experiences, a survey conducted in Germany with about 3000 students was able to show that the alienation from nature is clearly progressing compared to preceding surveys [48]. Many children no longer have any connection to nature, which generally seems to become more and more abstract for them. They know rules like "don't throw trash in nature", but cannot name in which compass direction the sun rises, for example [48]. The highest connection to nature is observed for children between seven and twelve years of age, while it is the lowest during the teenage years. In adulthood, the connection becomes stronger again [43]. With regard to attitudes toward the environment, some studies show that positive experiences in nature have a positive impact [43, 49–51]. Especially positive nature experiences in childhood and adolescence stimulate people to get involved in environment and nature later in life [43, 52]. More specifically, spending time in nature is a predictor of pro-environmental attitude towards nature and emotional connectedness with nature [43, 53, 54]. However, there is no information on how much time children should minimally spend in nature to improve their connection with it or their environmental attitude.

In view of the pressing environmental and related political challenges around the world, one might expect our children and youths to have a positive environmental attitude and AMBTS. However, actual assessment of this is still limited. In addition, studies are sparse regarding the question whether time spent in nature (TSIN) is related to environmental attitude and AMBTS. This leads to the following research questions:

What are the current environmental attitude (UTL and PRE) and AMBTS in secondary school students in Germany and do they differ in terms of gender, age, school type and SES?

How do environmental attitude (UTL and PRE) and AMBTS differ among students as a function of their TSIN?

## Materials and methods

### Study design and sample

Questionnaires to evaluate environmental attitude and AMBTS were distributed in German secondary schools in the Bavarian cities Munich and Augsburg. The study was conducted in accordance with the Declaration of Helsinki, and approved by the Ethics Committee of the Technical University of Munich (protocol code 505/19 S-SR; 02/20/2020). In total, 604 students presented written consent from parents or legal guardians to participate in the study and received the paper-pencil questionnaires. We used a 4- and a 5-point Likert scale for AMBTS and environmental attitude, respectively. These unequal scales result from the fact that the original scaled and validated response items were retained. Four hundred and eighty-four students filled them in during physical education classes, thus, the response rate was 80.1%. Of the participating students, 68.1% were female and 30.6% were male. The remaining 1.2% reported a diverse gender or did not deliver a gender statement. This group will be excluded in

the gender-specific analysis. Mean age was 12.08 ± .83 years, minimum age was 11 years and maximum age was 14 years. The Gymnasium formally represents the highest level of secondary school education in Germany (hereinafter referred to as higher secondary school (HSS)). The Realschule formally represents the second highest level of secondary school education (hereinafter referred to as lower secondary school (LSS)). Two hundred and forty-eight students from LSS (51.3%) and 235 from HSS (48.7%) took part. Mean international socio-economic index (ISEI) [55] as an indicator of the SES of the household where children grow up was 54.64 with a minimum of 15 and a maximum of 89. The ISEI is based on the current occupations of the parents. The mean value of the present sample corresponds to parents working as, for example, police officers or teachers.

## Measurements

**The 2-major environmental values model.**   Environmental attitudes are complex to measure and therefore require a psychometrically reliable and universally valid measurement instrument [38]. The 2-Major Environmental Values (2-MEV) scale (S1 File) for adolescents of Bogner and Wiseman [34] was developed in the 1990s, when scales for adults already existed.

Environmental attitude was measured using the UTL and PRE subscales of the 2-MEV. Participants responded to the questionnaire items by use of a 5-point Likert scale from 1 –"completely wrong" to 5 –"completely right". An example item for UTL is "Our planet has unlimited resources." An exemplary item for PRE is: "Humans are not more important than other living beings." The scales were used in their shortest version (UTL: 7 items; PRE: 6 items). Internal consistency was not satisfactory (Cronbach's alpha: UTL = .60; PRE = .42), which is probably due to the fact that the respective attitudes are assessed by highly diverse items that cover a wide range of aspects on which a given person might not have consistent thoughts and opinions. These considerations might apply even more strongly to the age group sampled for this study. A factor analysis, however, clearly indicated a two-dimensional structure with regard to high item loadings (> .50) on the respectively assumed latent factor and low cross-loadings (< .30).

**Affective-motivational beliefs towards sustainability.**   The subscale "affective-motivational beliefs towards sustainability" is one of six subscales in a sustainability competence questionnaire and has its origin in the scales of environmental attitude [35]. Waltner's response scale (S2 File) is a 4-point Likert scale from 1 - "disagree" to 4 - "agree." Someone who has positive AMBTS has high values in this case. The scale consists of 16 items, for example, "I think it's important to work for a just society". In our data, Cronbach's alpha of the used subscale was 0.81.

**Time spent in nature.**   To find out how much time students usually spend in nature, i.e. in green, natural surroundings like parks or forests, we asked them one question ("How many hours per week do you spend in nature?") and provided response categories to facilitate answers. There were four possible categories: "1 –half an hour or less"; "2 –between half an hour and two hours"; "3 –between two and three and a half hours" and "4 –three and a half hours or more".

**Socio-economic status.**   To determine the SES of the households in which students lived, the occupations of both parents and caregivers were queried by questionnaire. These were manually classified according to the International Standard Classification of Occupations (ISCO-08) system. The four-digit ISCO values were then converted to a two-digit ISEI-08 value. These values can range from 15 to 89. For example, a medical assistant with an ISCO value of 3256 is converted to an ISEI value of 46 [55]. The two-digit values make the

classification clearer. Higher ISEI values represent a higher SES. An ISEI value was only determined when the student could clearly indicate the occupation of at least one of his/her parents or legal guardians.

## Statistical analysis

For our statistical analysis, we used IBM SPSS statistics.

To separately analyze the roles of gender, age, school type, SES and TSIN for students' environmental attitude (PRE and UTL) and AMBTS, we conducted univariate regression analyses. In addition, we added multiple regression analyses including all independent variables simultaneously.

## Results

### Environmental attitude (UTL and PRE) and AMBTS in secondary school students

Table 1 shows the mean values and standard deviations of environmental attitude (UTL and PRE) and AMBTS for the subgroups according to gender, age, school type and TSIN. S1–S4 Figs show the mean values for each subgroup.

### Univariate and multiple regression analyses regarding the roles of gender, age, school type, SES and TSIN for students' environmental attitude (PRE and UTL) and AMBTS

Table 2 presents the results of the univariate regression analyses (left) and multiple regression analyses (right) for UTL, PRE and AMBTS with regard to the predictor variables gender, age, school type, SES and TSIN. The variable TSIN was dummy-coded. Since the three subgroups who indicated to spend at least half an hour per week in nature hardly differed from each

**Table 1. Mean values and standard deviations of environmental attitude (UTL and PRE) and AMBTS according to gender, age, school type and TSIN.**

| Subgroup | n | UTL | PRE | AMBTS |
|---|---|---|---|---|
| female | 328 | 2.04 (.57) | 3.94 (.62) | 3.42 (.45) |
| male | 148 | 2.03 (.65) | 4.02 (.54) | 3.27 (.58) |
| Age 11 | 127 | 2.00 (.49) | 4.03 (.55) | 3.44 (.37) |
| Age 12 | 206 | 2.02 (.59) | 3.95 (.56) | 3.40 (.48) |
| Age 13 | 126 | 2.03 (.61) | 3.95 (.61) | 3.31 (.52) |
| Age 14 | 19 | 2.51 (1.08) | 3.86 (.74) | 2.88 (.82) |
| LSS | 247 | 2.20 (.68) | 3.89 (.65) | 3.28 (.59) |
| HSS | 235 | 1.88 (.48) | 4.04 (.51) | 3.46 (.38) |
| TSIN—1 | 100 | 2.29 (.67) | 3.79 (.66) | 3.18 (.61) |
| TSIN—2 | 166 | 1.95 (.48) | 4.00 (.55) | 3.43 (.41) |
| TSIN—3 | 98 | 1.95 (.49) | 3.98 (.51) | 3.44 (.38) |
| TSIN—4 | 107 | 2.05 (.76) | 4.07 (.62) | 3.36 (.60) |

Note. N, number of participants in the respective subgroup; UTL, utilization; PRE, preservation; AMBTS, affective-motivational beliefs towards sustainability; TSIN, time spent in nature; LSS, lower secondary school; HSS, higher secondary school. The number after TSIN represents the item response categories regarding the amount of time per week spent in nature: 1, half an hour or less; 2, between half an hour and two hours; 3, between two and three and a half hours; 4, three and a half hours or more. Standard deviations are indicated in parentheses.

**Table 2. Results of univariate (left) and multiple (right) regression analyses examining the roles of gender, age, school type, SES and TSIN for UTL, PRE and AMBTS.**

| | Univariate regression analysis | | | | | Multiple regression analysis | | | |
|---|---|---|---|---|---|---|---|---|---|
| | **B** | **β** | **T** | **p** | **$R^2$** | **B** | **β** | **T** | **p** |
| **UTL** | | | | | | | | | |
| Gender | -.01 | -.01 | -.17 | .87 | - | .10 | .08 | 1.61 | .11 |
| **PRE** | | | | | | | | | |
| Gender | .09 | .07 | 1.46 | .15 | - | .03 | .02 | .46 | .65 |
| **AMBTS** | | | | | | | | | |
| Gender | -.15 | -.14 | -2.97 | < .01 | .02 | -.21 | -.20 | -4.21 | < .001 |
| | **B** | **β** | **T** | **p** | **$R^2$** | **B** | **β** | **T** | **p** |
| **UTL** | | | | | | | | | |
| Age | .07 | .09 | 2.02 | < .05 | .01 | .00 | .01 | .11 | .91 |
| **PRE** | | | | | | | | | |
| Age | -.04 | -.06 | -1.29 | .20 | - | -.03 | -.05 | -.99 | .33 |
| **AMBTS** | | | | | | | | | |
| Age | -.11 | -.18 | -3.96 | < .001 | .03 | -.05 | -.09 | -1.82 | .07 |
| | **B** | **β** | **T** | **p** | **$R^2$** | **B** | **β** | **T** | **p** |
| **UTL** | | | | | | | | | |
| School type | -.32 | -.26 | -5.91 | < .001 | .07 | -.28 | -.24 | -4.95 | < .001 |
| **PRE** | | | | | | | | | |
| School type | .15 | .13 | 2.86 | < .01 | .02 | .14 | .12 | 2.43 | < .05 |
| **AMBTS** | | | | | | | | | |
| School type | .19 | .18 | 4.10 | < .001 | .03 | .15 | .15 | 3.10 | < .01 |
| | **B** | **β** | **T** | **p** | **$R^2$** | **B** | **β** | **T** | **p** |
| **UTL** | | | | | | | | | |
| SES | -.01 | -.18 | -3.83 | < .001 | .03 | -.00 | -.11 | -2.19 | < .05 |
| **PRE** | | | | | | | | | |
| SES | .00 | .08 | 1.67 | .10 | - | .00 | .02 | .40 | .69 |
| **AMBTS** | | | | | | | | | |
| SES | .01 | .20 | 4.13 | < .001 | .04 | .00 | .14 | 2.98 | < .01 |
| | **B** | **β** | **T** | **p** | **$R^2$** | **B** | **β** | **T** | **p** |
| **UTL** | | | | | | | | | |
| TSIN D2 | -.34 | -.27 | -4.53 | < .001 | .04 | -.24 | -.19 | -3.06 | < .01 |
| TSIN D3 | -.35 | -.23 | -4.10 | < .001 | | -.19 | -.14 | -2.20 | < .05 |
| TSIN D4 | -.24 | -.17 | -2.89 | < .01 | | -.17 | -.12 | -1.96 | .05 |
| **PRE** | | | | | | | | | |
| TSIN D2 | .21 | .17 | 2.87 | < .01 | .02 | .14 | .12 | 1.78 | .08 |
| TSIN D3 | .20 | .14 | 2.37 | < .05 | | .05 | .03 | .52 | .60 |
| TSIN D4 | .28 | .20 | 3.48 | < .001 | | .23 | .17 | 2.61 | < .01 |
| **AMBTS** | | | | | | | | | |

(*Continued*)

**Table 2.** (Continued)

|  | Univariate regression analysis | | | | | Multiple regression analysis | | | |
|---|---|---|---|---|---|---|---|---|---|
|  | B | β | T | p | R² | B | β | T | p |
| TSIN D2 | .25 | .24 | 3.95 | < .001 | .03 | .20 | .20 | 3.14 | < .01 |
| TSIN D3 | .25 | .20 | 3.57 | < .001 |  | .19 | .16 | 2.58 | < .05 |
| TSIN D4 | .18 | .15 | 2.57 | < .05 |  | .15 | .13 | 2.13 | < .05 |

Note. UTL, utilization; PRE, preservation; AMBTS, affective-motivational beliefs towards sustainability; SES, socio-economic status; TSIN, time spent in nature; TSIN D2 –TSIN D4, dummy variables for item response categories 2 through 4; B, non-standardized regression coefficient; β, standardized regression coefficient; T, T value; p, p value; R², coefficient of determination for univariate regressions. Codings: Gender: female, 1; male, 2; school type: LSS, 1; HSS, 2.

other in their environmental attitude and AMBTS, but exhibited an obvious gap to the students who indicated to spend half an our per week in nature at the most (Table 1), item response category 1 was used as a reference category when examing the role of TSIN in regression analysis.

Univariate regression analyses suggest that attending a higher formal school type and spending more than half an hour per week in nature is positively associated with environmental attitude and AMBTS. Furthermore, utilization seems to increase with age, whereas SES exhibits a negative association to utilization. In line with this, results indicate that AMBTS become more negative with increasing age, whereas they are positively related to SES. Finally, compared to female students, male students seem to have more negative AMBTS.

Multiple regression analyses indicate that attending a higher formal school type, having a higher SES and spending more than half an hour per week in nature are significantly negatively related to UTL. The predictors explained 10.4% of the sample's variance in UTL.

Explaining 3.1% of the variance in PRE, attending a higher formal school type and spending at least three and a half hours per week in nature (compared to half an hour or less) were positively related to PRE.

Except for age, every variable included in the multiple regression analysis was a statistically significant predictor of AMBTS. A higher school type, higher SES and spending more than half an hour per week in nature were positive predictors. Additionally, compared to female students, male students exhibited less positive AMBTS. The model explained 11.7% of the variance in AMBTS.

## Discussion

With this study, we analyzed the environmental attitude (UTL and PRE) and AMBTS in secondary school students and their associations with gender, age, school type, SES and TSIN. In general, compared to other studies, the students' values in PRE and UTL represent a rather positive environmental attitude [37, 38]. Regarding AMBTS, the mean value of the entire sample is high as well [35]. Looking at the associations with gender, age, school type and SES, however, different needs for topic-related education emerged in specific sub-groups.

Regarding gender, both univariate and multiple regression analyses indicated that only AMBTS differed between female and male students, with female students exhibiting slightly higher values. This is in line with previous findings. According to several studies, there is a gender gap in favor of females in constructs of sustainability in Western countries [37, 56, 57]. Additionally, females show more positive attitudes in issues concerning solidarity and equity [46, 58], which are part of the construct of sustainability [14]. A similar gender gap has been found for environmental attitude measured with a 45-item scale [59]. Boewe-de Pauw et al. [57] and Olsson et al. [47] assume that girls are more positively influenced by the discussion of

environmental and sustainability topics in school than boys. Another reason could be that moralistic attitudes towards the environment are more pronounced in females [40]. In our results, however, we did not detect any gender differences in PRE or UTL. One explanation might be that moral aspects, in contrast to the scale used to assess AMBTS, are not sufficiently addressed in the scales that were used to assess environmental attitude.

Univariate regression analysis indicated that values in UTL increase and values in AMBTS decrease with increasing age of the students. This means that as the adolescents get older, exploitation of the environment might become more acceptable and their beliefs towards sustainability less strong. These findings are in line with existing literature from an environmental and sustainability point of view [35, 37, 41, 43, 60]. A possible explanation could be other factors that more strongly affect attitude and behavior during puberty. For example, a study that examined consumption behavior of young people showed that young people preferred a low price rather than focusing on the sustainability of an item of clothing [61]. This may be because consumption, e.g. current fashion, becomes more important at this age. As young people usually have a small amount of money at their disposal, they might be more likely to choose several unsustainable items rather than one that is sustainable, and therefore has a higher price [61]. However, explanations for the effect should be explored in future studies. Interestingly, the PRE values show no significant correlation with age. This is not in line with UTL and AMBTS. It corroborates the approach to consider environmental attitude as a two-dimensional construct [33, 34], and probably indicates that adolescents want nature to be preserved, irrespective of age. However, at some point and through various factors, as the one discussed above, their own interests seem to come first, which can lead to different developments in PRE and UTL within the same person. However, although the univariate regression analysis revealed minor trends with respect to age, it should be noted that only the underrepresented age group of 14-year-olds (see Table 1) differed substantially from the other age groups, which is why these regression analysis should be interpreted cautiously. This is reinforced by the results of the multiple regression analysis, which does not identify age as a predictor of UTL, PRE or AMBTS. Therefore, at least based on our data, the role of age can only be evaluated with reservation. However, there is potential that with a larger sample, a stronger trend or even significance would be evident. Baierl and colleagues [60], for example, could show in a sample of almost 2,000 adolescents that environmental attitudes decreased between 11/12 (maximum) and 16 years of age (minimum).

With regard to SES, higher ISEI values correlated with a more positive attitude towards the environment (lower values in UTL) and the AMBTS. This was underlined by the multiple regression analysis, which indicated SES to be a negative predictor of UTL and a positive predictor of AMBTS. The ISEI value depends on the parents' occupation, which, in turn, is associated with the educational level of the parents and thus the educational level of the household in which the child grows up. This suggests that children living in a household with a higher SES may be able to come to a better understanding and develop more differentiated views of environment and sustainability topics.

Previous studies show that higher education levels are associated with more pro-environmental attitudes [42, 44]. These findings can be supported with our data, for both environmental attitude and AMBTS. In multiple regression analyses, school type emerged as a predictor of unique incremental value above all other independent variables with respect to UTL, PRE and AMBTS. This finding is even more remarkable given the strong associations between school type and SES in Germany. On average, students attending HSS lived in households with a significantly higher ISEI than students attending LSS (HSS: mean = 58.4, LSS: mean = 50.76, t (433) = 4.694, p < .001). Thus, a somewhat higher standard of education is evident among the parents of the students in HSS. This could imply that the parents themselves already have a

more positive environmental attitude and higher AMBTS and probably transfer these values and beliefs to their children. Still, the results indicate that there must be independent aspects of school type itself that explain the differences that were revealed between children attending different school types. For example, it can be assumed that students at a HSS generally exhibit better learning abilities and may also be able to concentrate better [62, 63]. Thus, they might be better able to absorb the content that is taught, such as the content on environment and sustainability prescribed in the curriculum. Likewise, it is possible that teachers at HSS have more time to address advanced topics so that students have more elaborate experiences [62, 63].

The analyses on the assumed association between TSIN and environmental attitude and AMBTS yielded interesting results. Univariate regression analyses indicated that only the students who spent half an hour or less in nature per week have a more negative attitude towards the environment and worse AMBTS than all other students have. Based on this finding, a tentative recommendation could be made that students should spend at least 30 minutes per week in nature to establish more positive attitudes toward the environment and AMBTS. This is a very interesting finding, as it would imply a relatively low benchmark compared to other outcomes of TSIN. For benefits regarding health and well-being, for example, it is recommended to spend 120 minutes in nature per week [64]. At this point, however, it should be noted that our findings do not allow for definite conclusions regarding causality. One might argue that the students already have a more positive attitude towards nature and sustainability due to other reasons, e.g. a better education and/or a higher SES, and therefore spend more time in nature. However, the results of the multiple regression analyses on the association of TSIN with environmental attitude and AMBTS largely corroborate the results of the univariate regression analyses and thus suggest a unique role of TSIN that is independent of the school type that students attend and the SES of the household they live in. This finding may lend some further legitimacy to the assumption that at least 30 minutes per week in nature may promote a positive environmental attitude and AMBTS. This would also support previous studies that have identified a beneficial impact of positive experiences in nature on attitudes toward the environment. These studies, however, also point to the aspect that not only the amount of time [51] that is spent in nature but also the quality of the experience plays a role with regard to the environmental attitude and AMBTS. Richardson and colleagues [51], for example, found out that TSIN is not a significant predictor for pro-environmental behavior in adults. Instead, their data indicated that it is more important what is done in nature than how much time is spent in nature. They enumerate simple senses, such as smelling, seeing, and hearing in nature, that lead to a higher connectedness with nature, which was shown to be related with pro-environmental attitude [51, 65]. In any case, future studies that want to examine the association of environmental attitudes and AMBTS with nature experiences in young people should consider the amount of time that is spent in nature, the type of activities that are undertaken in nature and how enjoyable they were for the respective person.

The main strengths of this paper refer to the explanation of differences in environmental and sustainability attitudes of children living in Germany. While there is already some literature on gender and age effects, which we could use for comparisons, we could present new findings about the role of SES and school type with respect to environmental attitude and AMBTS. Generally, whereas associations of these explanatory factors with environmental attitude have been examined in some studies, only one previous study focused on AMBTS. From our point of view, our findings represent an exciting benefit, which allows more comparisons in the future. Regarding the question how much time children should regularly spend in nature to establish positive attitudes regarding the environment and sustainability, so far no satisfactory answers have been presented from a scientific perspective. Based on our results, we

refrained from examining the role of TSIN in isolated manner, and considered the amount of TSIN against the background of socio-economic and educational aspects.

This study also has some limitations that should be considered when interpreting the findings. Firstly, sustainability attitude is mostly considered as one part of sustainability competence, with the other parts addressing cognitive and behavioral aspects. We concentrated on one aspect of sustainability competence. The subscale "affective-motivational beliefs towards sustainability" led to values comparable with the original validation study. Further studies should consider investigating the cognitive and behavioral sub-dimensions of sustainability competence as well as sustainability competence as a whole. Secondly, internal consistency for the environmental attitude scales was low. This could be due to the fact that the questionnaire items cover a wide range of subjects, which makes high correlations between items less likely. Furthermore, the formation of attitudes of individuals in the selected age group is normally far from complete, which also means that their attitudes are not yet internally consistent. This, quite logically, is detrimental to internal consistency. A factor analysis of the 2-MEV scale with our sample showed that it assesses a two-dimensional construct with the PRE and UTL items each loading on separate factors. However, since UTL and PRE were predicted by different factors, the multiple regression analyses underlined the results of bivariate analyses showing that assessing data for both dimensions turned out to be an advantage. This, in sum, led us to assume that the instrument, despite its weak internal consistency, is suitable to capture environmental attitude in children and adolescents. Thirdly, our sample only included participants living in cities. Thus, in order to gain further insights into the addressed attitudes and their relation to time spent in green spaces, future studies should be conducted in urban but also in rural areas, where green spaces, such as parks or forests, may be easier to find and thus might be visited more often. Fourthly, we conducted our surveys only at two of the four types of schools that are common in Germany. It would be interesting to sample students from more school types. Additionally, it should be considered whether the curricula and excursions of the different types of schools have an influence on environmental attitude and AMBTS in students.

## Conclusions

In conclusion, we found that environmental attitude and AMBTS in higher secondary schools and lower secondary schools in Bavarian cities are on a relatively high level compared to other studies. Nevertheless, our data shows that a greater need for topic-related education seems to exist for adolescents with a lower SES and a lower formal level of education. To spend more than half an hour in nature might be helpful to increase a positive attitude for environmental and sustainability topics. AMBTS, as a part of sustainability competence, need a larger study base. Longitudinal studies on sustainability competence with affective, cognitive or behavioral priorities would be desirable because scientific studies regarding these aspects are limited. In addition, it would be exciting to formulate further research questions to investigate which nature activities might be of incremental value over and above socio-economic and educational aspects for strengthening students' pro-environmental attitude. In an effort to support these activities, it would be desirable to promote studies and institutions focused on education outside the classroom.

## Supporting information

**S1 Fig. Graphical representation of mean values according to school type.** UTL, utilization; PRE, preservation; AMBTS, affective-motivational beliefs towards sustainability; LSS, lower

secondary school; HSS, higher secondary school.
(PDF)

**S2 Fig. Graphical representaion of mean values according to gender.** UTL, utilization; PRE, preservation; AMBTS, affective-motivational beliefs towards sustainability.
(PDF)

**S3 Fig. Graphical representaion of mean values according to age.** UTL, utilization; PRE, preservation; AMBTS, affective-motivational beliefs towards sustainability.
(PDF)

**S4 Fig. Graphical representaion of mean values according to time spent in nature.** UTL, utilization; PRE, preservation; AMBTS, affective-motivational beliefs towards sustainability; TSIN, time spent in nature. The number after TSIN represents the item response categories regarding the amount of time per week spent in nature: 1, half an hour or less; 2, between half an hour and two hours; 3, between two and three and a half hours; 4, three and a half hours or more.
(PDF)

**S1 File. 2-Major Environmental Values (2-MEV) scale.**
(PDF)

**S2 File. Affective-motivational beliefs towards sustainability scale.**
(PDF)

**S1 Dataset. Dataset with dummy-variables.**
(XLSX)

## Acknowledgments

We thank all teachers and students, who were involved in this study. Additionally, we thank Anna Schulten (John Innes Centre, UK) for language editing.

## Author Contributions

**Conceptualization:** Charlotta Bucht, Joachim Bachner, Sarah Spengler.

**Data curation:** Charlotta Bucht.

**Formal analysis:** Charlotta Bucht, Joachim Bachner, Sarah Spengler.

**Funding acquisition:** Sarah Spengler.

**Investigation:** Charlotta Bucht.

**Methodology:** Charlotta Bucht, Joachim Bachner.

**Project administration:** Sarah Spengler.

**Supervision:** Joachim Bachner, Sarah Spengler.

**Visualization:** Charlotta Bucht.

**Writing – original draft:** Charlotta Bucht.

**Writing – review & editing:** Joachim Bachner, Sarah Spengler.

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
