## [Decision Letter · Decision Letter 0]

13 Jun 2023

PONE-D-23-04963Environmental attitude and affective-motivational beliefs towards sustainability of secondary school children in Germany and their associations with gender, age, school type, socio-economic status and time spent in naturePLOS ONE

Dear Dr. Bucht,

Thank you for submitting your manuscript to PLOS ONE. After careful consideration, we feel that it has merit but does not fully meet PLOS ONE’s publication criteria as it currently stands. Therefore, we invite you to submit a revised version of the manuscript that addresses the points raised during the review process.

We look forward to receiving your revised manuscript.

Kind regards,

Zakari Ali, PhD.

Academic Editor

PLOS ONE

Additional Editor Comments:

We apologise for delays in getting reviewer feedback to you. We have now obtained the required number of reviews for your manuscript and invite you to revise addressing concerns from the reviewers.

Reviewers' comments:

Reviewer's Responses to Questions

**Comments to the Author**

1. Is the manuscript technically sound, and do the data support the conclusions?

Reviewer #1: Yes

Reviewer #2: Yes

2. Has the statistical analysis been performed appropriately and rigorously? 

Reviewer #1: No

Reviewer #2: Yes

3. Have the authors made all data underlying the findings in their manuscript fully available?

Reviewer #1: No

Reviewer #2: Yes

4. Is the manuscript presented in an intelligible fashion and written in standard English?

Reviewer #1: Yes

Reviewer #2: Yes

5. Review Comments to the Author

Reviewer #1: Review of the article „Environmental attitude and affective-motivational beliefs towards sustainability of secondary school children in Germany and their associations with gender, age, school type, socioeconomic status and time spent in nature”

The article addresses an interesting and relevant topic. It describes sustainability and environmental attitudes in children and adolescents in Germany and investigates associations with sociodemographic characteristics and time spent in nature. Socioeconomic status and education are suggested to mainly shape these attitudes in children and adolescents.

The article is well written. However, theoretical aspects might be shortened, while descriptions of the outcomes in the sample should be more detailed. Furthermore, the analyses could be homogenized.

General comment:

You write at several point in the manuscript that your findings suggest a tentative recommendation to spend at least half an hour per week in nature. I do not believe that this would improve sustainability and environmental attitude. As you write in the Discussion, it is more probable that sustainability/environmental attitudes and spending time outdoors are both affected by a third factor, e.g., socioeconomic status of the family. Therefore, I would completely remove recommendations regarding time spending in nature.

Please avoid statements on causality (e.g., abstract, line 25, “predictor”).

For someone being no expert in the assessment of sustainability and environmental attitude, it is difficult to understand was actually assessed and how the students behaved. Could you provide a copy of the questionnaire items in the supplement? And maybe a figure on findings in your sample (not only means)?

In some sentences, the English does not seem to be correct. Can a native speaker check the manuscript?

Abstract:

I do not understand the last sentence (lines 34ff). I would remove it.

Introduction:

Line 42f: I suggest removing “[…] and created well understandable graphics”. This is not really relevant, is it?

Line 82: What is meant by “Knowledge about a certain object alone”? Which object?

Regarding environmental attitude, you mention different categorizations, e.g., “functions and structures“ (line 88) and “utilization and preservation” (line 96f). This is misleading.

Line 104ff: I suggest starting this paragraph with the term “affective-motivational beliefs” as part of sustainability attitudes (not with the term “sustainability attitudes”) as this is the variable you are assessing. Otherwise, it is misleading.

The introduction is too long. I suggest shortening explanations on the constructs. Rather give concrete examples.

Materials and methods:

Instead of using the term Gymnasium and Realschule (which are only known in Germany) please use a more general term, e.g., “higher” versus “lower” secondary school education. Also, you don’ have to mention that there exist four types of secondary school (line 170). As you did only distinguish two, information on 2 others in misleading.

Please provide percentages for children from lower and higher secondary schools.

Why are there so much more girl than boys in your sample?

Line 192f: Please provide values/ranges for high loading on the latent factor and for the low cross-loading.

Why don’t you perform linear regressions only? Why do you apply correlations, ANOVAs, and regressions? And sometimes ANOVAs AND linear regression? Using (only) regressions also for univariate associations would make results more comparable. Also, regression coefficients from uni- and multivariate analyses could be compared more easily.

Did you report standardized or non-standardized regression coefficients?

Results:

Table 1: Please add the possible range. This makes it easier to interpret the mean values. What does the last line of the Table tells us? Is it necessary? Shouldn’t it be the same as in Table 2 (which is not the case)? Please explain UTL and PRE in the caption. What do the * mean? What was compared (using which analyses)?

Please remove subsection headings 3.2.1-3.2.4.

Line 281f: time spend in nature is not really metric. However, it seems as if you included it as a metric independent variable into the regression. I think, that’s not the best way to analyze these data.

Discussion

Line 336f: Regarding age, you performed a correlation, didn’t you? Here, it seems as if you compared age groups. This is misleading.

Lines 419ff: The paragraph on internal consistency and factor loading is too long. These details are not needed in the limitations section.

Figure

Is this figure really necessary? A Figure on the outcomes would be more appropriate.

Reviewer #2: My first reaction with the paper was positive, when reading the harvest message, the sample sizes and the application of established scales. Many usual mistakes of studies in our field seemed omitted. However when reading a second, a third time, inconsistencies appeared. In consequence, the manuscript seems in a pre-mature stage and needs substantial improvement before accepting to publication in Plos1.

Let me go more into detail by limiting myqueries to 10 issues:

1) The sample’s age distribution is quite unbalanced (and needs discussion)

2) Table 2: The item itself is quite simple and maybe not valid, as time ranges for that age group might be overrated. Additionally, a age correction as covariate should be taken into consideration.

3) Knowledge, the manuscript is citing #31, but not differentiating the types of knowledge. The authors seemingly just concentrate on factual knowledge or system knowledge. A related application study of that specific model was published 2020 by Maurer & colleagues.

4) As the authors use 4- and 5-digit response patterns, this needs discussion.

5) The 3.4. sub-chapter needs more elaboration.

6) Conclusion: the first 2 sentences are just redundant

7) Paper 34 is insufficiently cited, therefore approaching that paper is not possible. In my vague memory from conferences, the cited scale of that group was not convincing.

8) Reference #60 seems to present an internal report rather than a peer-reviewed paper.

9) References are inconsistent anyway: Sometime with written first name, sometimes just with a letter (e.g. #60, 61). Some references seemingly were added to enrich the reference list as they are not integrated in argumentation lines.

10) The rooting in the literature body of the last two decades seems quite fragmentary: Especially two studies would have laid more foundation: Bogner & Wiseman (1997) [rural-urban samples] and Baierl et al (2022) [age gradient]. Especially for the latter a trend seems visible (which with higher samples sizes might have reached significance).

Nevertheless there is hope, my recommendation: Major revisions

6. PLOS authors have the option to publish the peer review history of their article (what does this mean?). If published, this will include your full peer review and any attached files.

Reviewer #1: No

Reviewer #2: No

---

## [Author Response · Author response to Decision Letter 0]

6 Aug 2023

Dear Editor, dear Reviewers,

We would like to thank you for your helpful comments. We believe that the quality of our manuscript has improved by implementing the suggestions and comments you made. Please find our response to the comments below. When we refer to lines, we always refer to the file labeled "Revised Manuscript with Track Changes". All our responses are written in italics. We hope we have addressed all your concerns and comments to your satisfaction.

Additional Editor Comments:

We apologise for delays in getting reviewer feedback to you. We have now obtained the required number of reviews for your manuscript and invite you to revise addressing concerns from the reviewers.

Reviewers' comments:

Reviewer's Responses to Questions

Comments to the Author

1. Is the manuscript technically sound, and do the data support the conclusions?

Reviewer #1: Yes

Reviewer #2: Yes

2. Has the statistical analysis been performed appropriately and rigorously? 

Reviewer #1: No

Reviewer #2: Yes

We substantially revised the statistical analysis according to the reviewers’ comments. We think that these changes benefited the quality of the findings and the discussion of the findings. Thank you again for your helpful suggestions.

3. Have the authors made all data underlying the findings in their manuscript fully available?

Reviewer #1: No

Reviewer #2: Yes

Thank you for your feedback. The data set (supplementary file) is included. Additionally, we now included the translated questionnaires. 

4. Is the manuscript presented in an intelligible fashion and written in standard English?

Reviewer #1: Yes

Reviewer #2: Yes

5. Review Comments to the Author

Reviewer #1: Review of the article „Environmental attitude and affective-motivational beliefs towards sustainability of secondary school children in Germany and their associations with gender, age, school type, socioeconomic status and time spent in nature”

The article addresses an interesting and relevant topic. It describes sustainability and environmental attitudes in children and adolescents in Germany and investigates associations with sociodemographic characteristics and time spent in nature. Socioeconomic status and education are suggested to mainly shape these attitudes in children and adolescents.

The article is well written. However, theoretical aspects might be shortened, while descriptions of the outcomes in the sample should be more detailed. Furthermore, the analyses could be homogenized.

General comment:

You write at several point in the manuscript that your findings suggest a tentative recommendation to spend at least half an hour per week in nature. I do not believe that this would improve sustainability and environmental attitude. As you write in the Discussion, it is more probable that sustainability/environmental attitudes and spending time outdoors are both affected by a third factor, e.g., socioeconomic status of the family. Therefore, I would completely remove recommendations regarding time spending in nature.

Thank you for your general comments. Based on the revision of our manuscript, we would like to address some of the general aspects that you mentioned here. In response to your suggestion, we homogenized the statistical analysis and now only perform linear regression analyses. As you mentioned in your comment, the variable ‘time spent in nature’ cannot really be considered metric. Therefore, it was dummy-coded in order to include it in the regression analysis. The multiple regression analysis indicated that time spent in nature does play a statistically significant role, also when socio-economic status and school type are simultaneously included in the analysis as further predictors. We would like to thank you again for your important recommendation regarding the statistical analysis since it has led to more accurate findings. Please see the results and discussion sections and our responses to your specific comments below for further information.

Please avoid statements on causality (e.g., abstract, line 25, “predictor”).

Thank you for this suggestion. Given our study design, we try to avoid to suggest any causal relationships and rather refer to terms like ‘associated to’ etc. wherever possible. The term ’predictor’ is only used within the framework of regression analyses to differentiate predictor and criterion variables.

For someone being no expert in the assessment of sustainability and environmental attitude, it is difficult to understand was actually assessed and how the students behaved. Could you provide a copy of the questionnaire items in the supplement? And maybe a figure on findings in your sample (not only means)?

Thank you for the helpful comment. We added our (translated) questionnaire as a supplementary file. The results section was largely revised (see also our responses to your other comments) and now includes tables where the results are visible at a glance. However, we were not sure what type of figures you had in mind. We have now added four figures as supporting file that illustrate the level of environmental attitude and affective-motivational beliefs towards sustainability in subgroups according to the different explanatory variables. If this is not what you had in mind when writing the comment, please let us know and we will add the figures accordingly.

In some sentences, the English does not seem to be correct. Can a native speaker check the manuscript?

Thank you very much for this suggestion. A native speaker colleague checked the language again. 

Abstract:

I do not understand the last sentence (lines 34ff). I would remove it. 

Thank you for this comment. We checked the sentence and removed it. 

Introduction:

Line 42f: I suggest removing “[…] and created well understandable graphics”. This is not really relevant, is it?

Thank you for this comment. We totally agree and deleted this part of the sentence. 

Line 82: What is meant by “Knowledge about a certain object alone”? Which object?

Thank you very much for your comment. We changed “certain object” into “environmental topics”. 

Regarding environmental attitude, you mention different categorizations, e.g., “functions and structures“ (line 88) and “utilization and preservation” (line 96f). This is misleading.

Thank you for your comment. We have rewritten the first sentence in this paragraph so that the misunderstanding does not occur in the text. Please see lines 99 - 100.

Line 104ff: I suggest starting this paragraph with the term “affective-motivational beliefs” as part of sustainability attitudes (not with the term “sustainability attitudes”) as this is the variable you are assessing. Otherwise, it is misleading.

Thank you for your comment. Unfortunately, we cannot implement your suggestion, since the beginning of the paragraph is about Attitude in general, and then follows the classification of affective-motivational beliefs towards sustainability. These are the closest to sustainability attitudes in the scientific context.

The introduction is too long. I suggest shortening explanations on the constructs. Rather give concrete examples.

Thank you very much for this comment. After discussing the idea of replacing the explanations of the constructs with concrete examples, we decided to not change the introduction. Understanding the constructs, what they refer to and what they do not refer to, is a rather complex task. Therefore, we find it very important to give elaborate explanations on the constructs to make sure that they are correctly understood. We hope that you are fine with this approach. 

Materials and methods:

Instead of using the term Gymnasium and Realschule (which are only known in Germany) please use a more general term, e.g., “higher” versus “lower” secondary school education. Also, you don’ have to mention that there exist four types of secondary school (line 170). As you did only distinguish two, information on 2 others in misleading.

Thank you for these comments. We changed the terms in the manuscript into higher and lower secondary schools (HSS/LSS) and deleted the sentence about the four school types in Germany.

Please provide percentages for children from lower and higher secondary schools.

Thank you for this comment. We added the percentages. See line 174.

Why are there so much more girl than boys in your sample?

In Bavaria, there are no coeducational physical education classes. Female teachers teach girls and male teachers teach boys. The female teachers were more motivated to take part in the study than the male teachers were. Accordingly, more data was collected from girls.

Line 192f: Please provide values/ranges for high loading on the latent factor and for the low cross-loading.

Thank you for this comment. We have reworded the sentence so that it can now be read out which loadings are present on the latent factor and for the cross-loading. Please see lines 210 - 212.

Why don’t you perform linear regressions only? Why do you apply correlations, ANOVAs, and regressions? And sometimes ANOVAs AND linear regression? Using (only) regressions also for univariate associations would make results more comparable. Also, regression coefficients from uni- and multivariate analyses could be compared more easily.

Thank you for this comment. We have completely revised the results section and made new calculations. You will now find univariate regression analyses in the results section in addition to the multiple regression analysis. We also calculated regressions for "time spent in nature" by creating dummy variables. Thus, the results are now more comparable.

Did you report standardized or non-standardized regression coefficients?

In the revised results section, we now report both standardized and non-standardized regression coefficients.

Results:

Table 1: Please add the possible range. This makes it easier to interpret the mean values. What does the last line of the Table tells us? Is it necessary? Shouldn’t it be the same as in Table 2 (which is not the case)? Please explain UTL and PRE in the caption. What do the * mean? What was compared (using which analyses)?

Thank you for the comments and questions. The range is described in the methods section (please see lines 202 – 205 and 218 - 219). We revised Table 1, so that you can see mean values and standard deviations. The last line of the table and the * were removed (not useful anymore due to the substantial revision of the methods section) and a note is added, where every abbreviation is explained. 

Please remove subsection headings 3.2.1-3.2.4.

Thank you for this comment. We removed the subsections. 

Line 281f: time spend in nature is not really metric. However, it seems as if you included it as a metric independent variable into the regression. I think, that’s not the best way to analyze these data.

Thank you for this comment. As mentioned earlier, we now use dummy variables for time spent in nature to be able to include the variable in linear regression analyses. 

Discussion

Line 336f: Regarding age, you performed a correlation, didn’t you? Here, it seems as if you compared age groups. This is misleading.

Thank you for your comment. We now refer to Table 1 in the text and have added a paragraph so that this misunderstanding no longer prevails. Please see lines 417 – 428.

Lines 419ff: The paragraph on internal consistency and factor loading is too long. These details are not needed in the limitations section.

Thank you for this comment. We agree that the statements on internal consistency and factor analysis is quite long. However, we consider these aspects to be of high importance when conducting a questionnaire assessment. Therefore, we wanted to discuss the limitation of low internal consistency in more detail and, most importantly, transparently.

Figure

Is this figure really necessary? A Figure on the outcomes would be more appropriate.

Thank you for your comment. As we wrote in our response to one of your comments above, we added four figures and removed the one referring to ISEI. 

 

Reviewer #2: My first reaction with the paper was positive, when reading the harvest message, the sample sizes and the application of established scales. Many usual mistakes of studies in our field seemed omitted. However when reading a second, a third time, inconsistencies appeared. In consequence, the manuscript seems in a pre-mature stage and needs substantial improvement before accepting to publication in Plos1.

Let me go more into detail by limiting my queries to 10 issues:

1) The sample’s age distribution is quite unbalanced (and needs discussion).

Thank you for this comment. We are aware of the unequal distribution and for this reason, we have written a section in the Discussion that refers precisely to this. Please see lines 417 – 428.

2) Table 2: The item itself is quite simple and maybe not valid, as time ranges for that age group might be overrated. Additionally, a age correction as covariate should be taken into consideration.

Thank you very much for this hint. We are aware of the possible inaccuracy of time estimates of adolescents and have therefore designed the categories rather broadly, so that at least the assignment to these broader categories is relatively reliable. In our opinion, a more reliable and at the same time economical and reasonable way of assessing the construct was not possible. Diaries would certainly be more reliable, but also more time-consuming, and thus less economical. Due to probably emerging compliance problems, the sample size with regard to this variable would then have been substantially reduced. 

Thank you for your suggestion regarding a potential age correction. Since we have conducted a multiple regression analysis, an age correction is automatically done since all predictor variables are included simultaneously.

3) Knowledge, the manuscript is citing #31, but not differentiating the types of knowledge. The authors seemingly just concentrate on factual knowledge or system knowledge. A related application study of that specific model was published 2020 by Maurer & colleagues.

Thank you for your comment. We have now included the differentiation of the different types of knowledge (please see lines 89 - 92). We subsequently define these in one sentence, as our article is exclusively about environmental attitude and we would like to mention knowledge and behavior only briefly in order to avoid any misunderstandings.

4) As the authors use 4- and 5-digit response patterns, this needs discussion.

Thank you for this comment. We added the information about it to the first paragraph in the Discussion section. “We used a 4- and a 5-point Likert scale for affective-motivational beliefs towards sustainability respectively environmental attitude. These unequal scales result from the fact that the original scaled and validated response items were retained.”

5) The 3.4. sub-chapter needs more elaboration.

Thank you for your comment. We added a table and revised the text in this paragraph. A more detailed discussion of these results can be found in the Discussion section; please see for example lines 422 - 426. 

6) Conclusion: the first 2 sentences are just redundant

Thank you for this comment. The first sentence summarizes the study that all values are higher compared to other studies. At the same time, however, a difference can be seen within our data: The scores of children with low socioeconomic status and lower formal level are lower than those of the other groups. Thus, there is a more intensive need for environmental education for these groups. We have tried to resolve this with formulations that are more precise. See lines 527 - 530.

7) Paper 34 is insufficiently cited, therefore approaching that paper is not possible. In my vague memory from conferences, the cited scale of that group was not convincing.

Thank you for this hint. We corrected the citation. 

In the article (Waltner et al., 2019), Cronbach’s alpha for the scale we used was .84 and the conclusion of the paper is “According to the presented results, the questionnaire can be approved for practical application to measure different dimensions of sustainability competencies”. When we started to plan and conduct our data assessment in 2019, there were no validated scales available to assess sustainability attitude or similar constructs. There were also no articles that criticized the scale or even recommended not to use it. This has not changed until today, and this despite the fact that, in the meantime, the validation study has been frequently cited.

8) Reference #60 seems to present an internal report rather than a peer-reviewed paper.

Thank you for this comment. It is a political report summarizing the results of the OECD Pisa study 2018. We added the official PISA 2018 Results for Germany as reference. The authors did not find a peer-reviewed paper in the context of concentration and learning experience in German schools. 

9) References are inconsistent anyway: Sometime with written first name, sometimes just with a letter (e.g. #60, 61). Some references seemingly were added to enrich the reference list as they are not integrated in argumentation lines.

Thank you for this comment. The reference list has been completely revised so that now only last name and the first letter of the first name appear. However, no references had been added to the reference list that would not have appeared in the main text. All references had been inserted in the mansucript via EndNote.

10) The rooting in the literature body of the last two decades seems quite fragmentary: Especially two studies would have laid more foundation: Bogner & Wiseman (1997) [rural-urban samples] and Baierl et al (2022) [age gradient]. Especially for the latter a trend seems visible (which with higher samples sizes might have reached significance).

Thank you very much for your comment and literature suggestions. Bogner and Wiseman (1997) is well known by the authors but we could not find a way to include the findings of this article. We did not conduct a rural-urban comparison and the study discusses verbal commitment and environmental behavior, which we did not address in our study. Baierl et al. (2022) fit into the discussion nicely.

Nevertheless there is hope, my recommendation: Major revisions

---

## [Decision Letter · Decision Letter 1]

25 Sep 2023

PONE-D-23-04963R1Environmental attitude and affective-motivational beliefs towards sustainability of secondary school children in Germany and their associations with gender, age, school type, socio-economic status and time spent in naturePLOS ONE

Dear Dr. Bucht,

Thank you for submitting your manuscript to PLOS ONE. After careful consideration, we feel that it has merit but does not fully meet PLOS ONE’s publication criteria as it currently stands. Therefore, we invite you to submit a revised version of the manuscript that addresses the points raised during the review process.

We look forward to receiving your revised manuscript.

Kind regards,

Zakari Ali, PhD.

Academic Editor

PLOS ONE

Journal Requirements:

Reviewers' comments:

Reviewer's Responses to Questions

**Comments to the Author**

1. If the authors have adequately addressed your comments raised in a previous round of review and you feel that this manuscript is now acceptable for publication, you may indicate that here to bypass the “Comments to the Author” section, enter your conflict of interest statement in the “Confidential to Editor” section, and submit your "Accept" recommendation.

Reviewer #1: (No Response)

Reviewer #2: (No Response)

2. Is the manuscript technically sound, and do the data support the conclusions?

Reviewer #1: Yes

Reviewer #2: Partly

3. Has the statistical analysis been performed appropriately and rigorously? 

Reviewer #1: Yes

Reviewer #2: Yes

4. Have the authors made all data underlying the findings in their manuscript fully available?

Reviewer #1: Yes

Reviewer #2: No

5. Is the manuscript presented in an intelligible fashion and written in standard English?

Reviewer #1: Yes

Reviewer #2: Yes

6. Review Comments to the Author

Reviewer #1: Thank you for addressing the comments I made. I have only a few minor comments:

1. Second line within the "Study design and sample" section: You now refer to high and lower secondary school, which is totally correct. Therefore, please remove "Realschule" and "Gymnasium" also at the beginning of this section. "Secondary school" is sufficieltn at that point.

2. I suggest combining Table 2 and Table 3 to one Table (this allows direct comparisons between univariate and multiple regression). For example, put information on Table 2 on the left, and information of Table 3 on the right.

3. In the tables, please make clear to which gendder/school type the numbers refer to (I think, it shoud be "male" instead of gender, and higher secondary school instead of school type).

4. You mention in the Discussion that TSIN was only significantly associated with PRE in multiple regression analysis. However, according to Table 3, the associations with UTL and affective-motivational beliefs towards sustainability were stronger (and significant). Please clarify.

Reviewer #2: 1) The sample’s age distribution is quite unbalanced (and needs discussion)

OK

2) Table 2: The item itself is quite simple and maybe not valid, as time ranges for that age group might be overrated. Additionally, a age correction as covariate should be taken into consideration.

OK

3) Knowledge, the manuscript is citing #31, but not differentiating the types of knowledge. The authors seemingly just concentrate on factual knowledge or system knowledge. A related application study of that specific model was published 2020 by Michaela Maurer and colleagues.

Still pending, but I understand that this is not repairable anymore. Thus OK if you acknowledge this as a limitation

minor: line 91 (Roczen and colleagues); line 92 (Kaiser and colleagues)

4) As the authors use 4- and 5-digit response patterns, this needs discussion.

OK

5) The 3.4. sub-chapter needs more elaboration

OK

6) Conclusion: the first 2 sentences are just redundant

Unclear response as in my pdf-version lines 527 – 530 points to reference 40 and 41

7) Paper 34 is insufficiently cited, therefore approaching that paper is not possible. In my vague memory form conferences, the cited scale of that group was not convincing.

Still insufficient, better drop this citation

8) Reference #60 seems to present an internal report rather than a peer-reviewed paper.

still insufficiently cited, just drop this citation as this is just an internal report

9) References are inconsistent anyway: Sometime with written first name, sometimes just with a letter (e.g. #60, 61). Some references seemingly were added to enrich the reference list as they are not integrated in argumentation lines

For the new #60, still incorrect, nevertheless I recommend dropping anyway (see 9)

10) The rooting in the literature body of the last two decades seems quite fragmentary: Especially two studies would have laid more foundation: Bogner & Wiseman (1997) [rural-urban samples] and Baierl et al (2022) [age gradient]. Especially for the latter a trend seems visible (which with higher samples sizes might have reached significance).

Agree with the first one, but couldn't locate “Baierl and colleagues” in pointing to age coefficients in the new version

11) New: Figure 2: A lot of space of presenting no difference. How about reducing to two subsample; 30 min or less vs 30 min or more (i.e. difference vs no difference)

Recommendation: Minor revisions

7. PLOS authors have the option to publish the peer review history of their article (what does this mean?). If published, this will include your full peer review and any attached files.

Reviewer #1: No

Reviewer #2: No

---

## [Author Response · Author response to Decision Letter 1]

17 Nov 2023

Dear Editor, dear Reviewers,

We would like to thank you again for your helpful comments. We believe that the quality of our manuscript has improved by implementing the suggestions and comments you made. Please find our response to the comments below. When we indicate line numbers, we always refer to the file labeled "Revised Manuscript with Track Changes". All our responses are written in italics. We hope we have addressed all your concerns and comments to your satisfaction.

Reviewer 1

Reviewer #1: Thank you for addressing the comments I made. I have only a few minor comments:

1. Second line within the "Study design and sample" section: You now refer to high and lower secondary school, which is totally correct. Therefore, please remove "Realschule" and "Gymnasium" also at the beginning of this section. "Secondary school" is sufficient at that point.

Thank you very much for this comment. We removed “Realschule and Gymnasium” in line 177. 

2. I suggest combining Table 2 and Table 3 to one Table (this allows direct comparisons between univariate and multiple regression). For example, put information on Table 2 on the left, and information of Table 3 on the right.

Thank you very much for this idea. We have combined Table 2 and Table 3 into one Table (now Table 2). The text that refers to the results presented in the table was slightly adapted accordingly.

3. In the tables, please make clear to which gender/school type the numbers refer to (I think, it should be "male" instead of gender, and higher secondary school instead of school type).

Thank you for your helpful comment. We translated your hint into another idea. Therefore, we used the notes under Table 2 to indicate how gender and school type were coded so that the readers can interpret the values more easily. This way we could name all predictors according to the general construct and did not have to mix construct names with indications of specific types (male, HSS) of the constructs.

4. You mention in the Discussion that TSIN was only significantly associated with PRE in multiple regression analysis. However, according to Table 3, the associations with UTL and affective-motivational beliefs towards sustainability were stronger (and significant). Please clarify.

Thank you very much for this comment. You are completely right. Thank you for bringing this error to our attention. We have corrected the respective part of the discussion and adapted it so that it matches the results, please see lines 403-425 and lines 430-434. Furthermore, we revised the abstract accordingly, please see lines 29-35.

Reviewer 2

Dear Reviewer,

When reading your comments, it became clear that you have apparently and unfortunately read and referred to the line specifications and content of the February 2023 version of our manuscript (the first version that was submitted). In reaction to your comments regarding the originally submitted version, we had addressed your concerns and comments in the penultimate version of our manuscript, which we submitted in August 2023. In the following, we now would like to respond to those comments again in which you (quite logically) expressed your dissatisfaction with the last revision of the manuscript.

Reviewer #2: 1) The sample’s age distribution is quite unbalanced (and needs discussion)

OK

2) Table 2: The item itself is quite simple and maybe not valid, as time ranges for that age group might be overrated. Additionally, a age correction as covariate should be taken into consideration.

OK

3) Knowledge, the manuscript is citing #31, but not differentiating the types of knowledge. The authors seemingly just concentrate on factual knowledge or system knowledge. A related application study of that specific model was published 2020 by Michaela Maurer and colleagues.

Still pending, but I understand that this is not repairable anymore. Thus OK if you acknowledge this as a limitation

Thank you for your comment. We had included the differentiation of the different types of knowledge (please see lines 91-94). However, we had subsequently defined these in only one sentence, as our article is exclusively about environmental attitude and we would like to mention knowledge and behavior only briefly in order to avoid any misunderstandings or lead the reader away from the core aspects of the article.

minor: line 91 (Roczen and colleagues); line 92 (Kaiser and colleagues)

Thank you for this comment. We have added “colleagues”. Please see lines 104 and 105.

4) As the authors use 4- and 5-digit response patterns, this needs discussion.

OK

5) The 3.4. sub-chapter needs more elaboration

OK

6) Conclusion: the first 2 sentences are just redundant

Unclear response as in my pdf-version lines 527 – 530 points to reference 40 and 41

Thank you for this comment. The first sentence of the conclusion refers to the fact that the values of our sample in the constructs of environmental attitude and affective-motivational beliefs towards sustainability tend to be more positive compared to other studies. The second sentence refers to the fact that, at the same time, subgroup-related differences can be seen within our data: The scores of children with low socioeconomic status and lower formal level of education are less positive (in a pro-environmental sense) than those of the other groups. Thus, there is a more intensive need for environmental education for these subgroups. In the penultimate version of the manuscript, we had tried to resolve this problem with formulations that are more precise. Please see lines 473-477 in the most current version.

7) Paper 34 is insufficiently cited, therefore approaching that paper is not possible. In my vague memory form conferences, the cited scale of that group was not convincing.

Still insufficient, better drop this citation

Thank you for this comment. We have revised the reference. However, we cannot drop the citation because it is the reference for the PRE and UTL scales that we have used.

8) Reference #60 seems to present an internal report rather than a peer-reviewed paper.

still insufficiently cited, just drop this citation as this is just an internal report

Thank you for this comment. We now have removed #62 “PISA 2018 Grundbildung im internationalen Vergleich – Zusammenfassung“ as we think that you refer to this reference (reference #62 in the current version was reference #60 in the original version that you have commented on).

9) References are inconsistent anyway: Sometime with written first name, sometimes just with a letter (e.g. #60, 61).

Some references seemingly were added to enrich the reference list as they are not integrated in argumentation lines

For the new #60, still incorrect, nevertheless I recommend dropping anyway (see 9)

Thank you for this comment. We dropped it. 

10) The rooting in the literature body of the last two decades seems quite fragmentary: Especially two studies would have laid more foundation: Bogner & Wiseman (1997) [rural-urban samples] and Baierl et al (2022) [age gradient]. Especially for the latter a trend seems visible (which with higher samples sizes might have reached significance).

Agree with the first one, but couldn't locate “Baierl and colleagues” in pointing to age coefficients in the new version

Thank you for this comment. We had referred to Baierl and colleagues in the penultimate version of the manuscript. Please see lines 360-362in the current version.

11) New: Figure 2: A lot of space of presenting no difference. How about reducing to two subsample; 30 min or less vs 30 min or more (i.e. difference vs no difference)

Thank you for this comment. In an earlier version of the manuscript, we had conducted the analysis roughly the way you suggest it to be done. However, in reaction to an advice of one reviewer, we have changed this and used dummy variables for time spent in nature in order to be able to include it in the regression analyses. We really appreciate the benefits that come with this type of analysis as it allows to examine whether variable levels 2 through 4 all differ from variable level 1 and whether there are differences between levels 2, 3 and 4 (when looking at the B values).

Apart from that, we have now combined the former version of Table 2 and Table 3 into one table (now Table 2) in order to save space and make comparisons of the results of the univariate and multiple regression analyses easier. This was a suggestion from Reviewer 1. We hope you also agree with this.

---

## [Editor Report · Decision Letter 2]

12 Dec 2023

Environmental attitude and affective-motivational beliefs towards sustainability of secondary school children in Germany and their associations with gender, age, school type, socio-economic status and time spent in nature

PONE-D-23-04963R2

Dear Dr. Bucht,

We’re pleased to inform you that your manuscript has been judged scientifically suitable for publication and will be formally accepted for publication once it meets all outstanding technical requirements.

Kind regards,

Zakari Ali, PhD.

Academic Editor

PLOS ONE
---

## [Editor Report · Acceptance letter]

5 Apr 2024

PONE-D-23-04963R2 

PLOS ONE

Dear Dr. Bucht, 

I'm pleased to inform you that your manuscript has been deemed suitable for publication in PLOS ONE. Congratulations! Your manuscript is now being handed over to our production team.

Kind regards, 

on behalf of

Dr. Zakari Ali 

Academic Editor

PLOS ONE